# Food Security: 3D Dynamic Display and Early Warning Platform Construction and Security Strategy

**DOI:** 10.3390/ijerph191811169

**Published:** 2022-09-06

**Authors:** Ning Sun, Sai Tang, Ju Zhang, Jiaxin Wu, Hongwei Wang

**Affiliations:** 1School of Humanities, Social Sciences and Law, Harbin Institute of Technology, Harbin 150001, China; 2International Relations Faculty, Al-Farabi Kazakh National University, Almaty 050000, Kazakhstan

**Keywords:** food security, 3D dynamic display, early warning platform, security strategy, SOM, SVR

## Abstract

Since it affects a nation’s economy and people’s wellbeing, food security is a crucial national security requirement. In order to realize multi-angle grain data presentation and analysis and achieve the goal of deep mining, we propose a 3D dynamic visualization analysis method of multidimensional agricultural spatial–temporal data based on the self-organizing map. This method realizes the multi-angle display and analysis of grain data and achieves the purpose of deep mining. With the outbreak of COVID-19, the global food security situation is not optimistic, so it is necessary to use the food security early warning system to solve the food security issue. Machine learning has emerged widely in recent years and has been applied in various fields. Therefore, it is an excellent way to solve food security to apply the model in machine learning to construct a food security early warning system. Afterward, a food security early warning platform is developed with a support vector regression (SVR) model to ensure food security. Finally, we analyze China’s medium and long-term food security policy in line with modernization objectives. The experimental results show that the food security early warning platform based on the SVR model from 2007 to 2016 is effective compared with the actual situation every year. Through analyses, we should improve the stability, reliability, and sustainability of food supply, firmly hold the food security initiative, and construct a national food security guarantee system matching the goal of modernization.

## 1. Introduction

Food is a tactical issue, while food security is a strategic one. Under the influence of unilateralism, trade conservatism, and anti-globalization trend, the average international food trade is facing challenges [1,2]. Domestic grain production is affected by land resources, agricultural science, technology, and the grain market and has also encountered a bottleneck period of development [3,4]. When uncertainties and destabilizing factors rise, we must maintain strategic focus and ensure good food production and national food security [5,6,7,8]. Food security refers to a fundamental human right to life, namely, that everyone should have access to enough food for future survival and health. Specifically, according to the Food and Agriculture Organization (FAO), food security refers to all people, at all times, having physical and economic access to sufficient, safe, and nutritious food that meets their dietary needs and food preferences for an active and healthy life [9,10,11,12]. At the national level, food security means that a country’s food production, supply, and sale are in a state of no danger and threat. Food security bears on political stability, national economy, and people’s livelihood and is an essential foundation for overall national security. According to the FAO, nearly one-third of all food produced for human consumption (1300 Mt of food) is lost or wasted every year [13]. One of the main challenges towards the enhancement of food security is the reduction of food waste. As a result of COVID-19, the lockdown in Italy has stimulated snack consumption, particularly potato chips. Therefore, Amicarelli et al., used material flow cost accounting to measure the economic costs of food loss and waste in the Italian potato chip industry for sustainable resource and waste management [14]. Poore and Nemecek investigated how food producers might reduce their environmental effects by monitoring and reporting their effects to customers [15]. In response to food insecurity, emergency response capacity should be increased. While grain production is of particular importance, to solve the problem of food security, it must be based on grain production, improve grain production capacity, and maintain a high level of self-sufficiency [16,17]. In recent years, some achievements have been made in researching food security early warning systems. Since the early warning system is enormous, all factors are interrelated, and there are fuzzy indicator standards and different classification concepts, the current food security early warning system is too simple, and there is no unified and practical effective system for food security early warning. Machine learning has emerged widely in recent years and applied to various fields [18]. Therefore, applying the model in machine learning to construct a food security early warning system is an excellent way to solve the current dilemma.

Food security is not only a global issue but also a national one. It is a comprehensive international and domestic issue. The 2021 edition of the state of food security and nutrition in the world report estimates that the excess number of hungry people associated with the COVID-19 pandemic will reach 30 million by the end of the century. In 2020, nearly 12% of the global population was severely in food insecurity, representing 928 million people and 148 million more than in 2019 [19]. At the global level, the incidence of moderate or severe and only severe food insecurity is higher in women than in men and more severe in rural areas [20]. While the global food security situation is deteriorating, the food security risk in China is also gradually prominent [21,22,23]. The apparent imbalance between supply and demand is an easy cause of the food crisis. China’s total grain output has steadily declined since 2000 and dropped to around 430 Mt in 2003. Since 2004, China’s total grain output has been a bumper harvest, entering an era of 13 consecutive years of growth. In 2017, China’s grain output reached 617.9 Mt. Urbanization has reduced the area of agricultural land. Compared with the grain area in 2017, the area decreased by 960,000 ha in 2018. Coupled with the frequent occurrence of extreme weather and various disasters, China’s grain output decreased to 657.89 Mt in 2018, and the trend will continue to decline [24,25]. The growth of total grain output has hit a bottleneck.

With the application of global positioning systems, remote sensing technology, the Internet of Things (IoT), and other agricultural production technologies, many multidimensional spatial–temporal data are generated [26,27,28]. This data effectively records and displays the development of things at various stages, a kind of high-dimensional data with complex structure, multi-layer nesting, and spatial and temporal characteristics. Due to the strong correlation of data in time and space, it contains excellent mining potential. How to mine and analyze this data is of great significance to the development of fine agriculture, agricultural production, and modern society. Rana et al., analyzed the literature and concluded that using IoT-supported blockchain technology contributes to the sustainability of agri-food production. However, this technique can lead to some challenges [29]. Bux et al., conducted a comprehensive literature review assessing the sustainability of halal food, examining the barriers and opportunities provided by authentication and blockchain tools [30]. Dimensional reduction mapping technology can project multidimensional or high-dimensional data into two or three-dimensional space, display the dataset’s clustering structure and data distribution by scattered graphs, and the clustering result from standard information [31]. Therefore, it is widely used in information visualization with high dimensions and extensive data. In this study, a self-organizing map (SOM) is used to reduce the dimension of high-dimensional data [32]. When the data are analyzed and expressed by other projection technologies, it can produce good projection results with a small calculation. Since a single visualization method cannot meet the requirements of multi-angle representation and analysis of spatial–temporal data, the integrated visualization method is only a combination of spatial–temporal data visualization tools. In essence, the spatial–temporal data are expressed and explained independently. This study proposed multiple views based on SOM collaborative visualization of 3D analysis method, this method from the perspective of visualization for the data dimension reduction, integration of a variety of visual analysis tools, data visualization expression after for dimension reduction, both to solve the traditional visual analysis tools cannot provide visualization of high dimensional multiple attribute data of time and space. Moreover, the linkage between various expression tools can realize the real-time multi-angle visual expression and data analysis and enhance analysts’ ability to mine confidential information, which is of great significance to the 3D dynamic display of food security.

Under the influence of the international environment, China’s food security situation has also been negatively affected. However, China attaches great importance to this aspect of work and makes it an important national strategy. The deterioration of the international food security situation makes China unable to stay immune [33,34]. In addition, as a significant food country, China attaches great importance to food security. Therefore, the research on food security, especially in the past, which is used to observe the current and future warnings, develops rapidly. Under the threat of global food security, it is significant to guarantee China’s food security and provide a guarantee for global food security [35,36]. The research on China’s early warning system can make judgments and predictions for the state of domestic food security. Under various uncertainties, it is not only of theoretical significance to guarantee China’s food security but also of practical significance to guide practical economic activities.

The contributions of this paper are summarized as follows. (i) SOM-based collaborative multi-view 3D dynamic visualization of food security is presented. (ii) The SVR-based food security early warning platform is constructed. (iii) The medium and long-term food security strategy is discussed.

The rest of the paper is structured as follows. In Section 2, we study the materials and methods of the construction of food security early warning platforms. Experimental results are reported in Section 3. Section 4 gives the discussion on medium and long-term food security strategy, and Section 5 concludes this paper.

## 2. Materials and Methods

This section presents a SOM-based collaborative multi-view 3D dynamic visualization of food security. This method uses SOM to reduce the dimension of multidimensional grain data, combines with a parallel coordinate system, space–time cube, and other visualization components, realizes multi-angle display and analysis of grain data, and achieves the purpose of deep mining. Most strikingly, we construct the SVR-based food security early warning platform for food data. The overall framework is shown in Figure 1.

### 2.1. Theoretical Background

As a classical visualization method of high dimensional data in the two-dimensional plane, the parallel coordinate represents geometric visualization technology [37]. Parallel coordinates can be able to visually express the relationship between data through projective geometric interpretation and dual features without using vectors or other visual icons and are easy to understand. The disadvantage is that the increase in data volume and the increase in polyline density led to many overlapping lines, which are difficult to identify [38,39]. The original grain data’s dimension is reduced via SOM-based collaborative multi-view 3D dynamic visualization of food security, which then visualizes the parallel coordinates. When using parallel coordinates to further supplement the link between multidimensional grain data, it may better avoid the drawbacks of a high number of overlapping lines brought on by increased data volume and broken line density. According to the clustering results obtained from SOM dimension reduction, typical data in each class are obtained as input variables. Referring to the visualization technology based on parallel coordinates such as data abstraction, coordinate axis exchange, and dimension control, parallel axes are set according to data attribute dimensions to realize the visual expression of spatial–temporal data, and corresponding colors are assigned to each class for convenient observation and analysis. SOM-based collaborative multi-view 3D dynamic visualization of food security provides data support for the following enhanced food security warning platform. Sood and Singh discussed how to reduce food waste by using computer vision and machine learning methods [40]. For sub-Saharan farmers who own very little farmland, Khalif and Nur discussed the role of African farmers in reducing food insecurity [41]. In the production of food, pesticides are commonly utilized. Brazil’s shortcomings in monitoring the use of pesticides have been examined by Gerage et al. In order to increase food security, sustainable agriculture must use fewer pesticides [42]. Farmland and water resources are the main factors that influence food production in India. Through the analyses, Kumar et al. found that the gap between agricultural water demand and water availability is the core of problems with both food security and water management. As a result, they proposed appropriate solutions to the water shortage [43].

A space–time cube is mainly used to express the space–time path [44,45]. The space–time cube model aggregates the sample points into the spatial–temporal data structure employing bin time series. By creating a space–time cube, spatial–temporal data can be visualized and analyzed in the form of time series analysis and integrated spatial and temporal pattern analysis, as shown in Figure 2. In Figure 2, the *x*-axis and *y*-axis represent the spatial position of the time, while the *z*-axis represents the time. The bottom layer is the start time, and the top layer is the latest time. Each cube is composed of the attribute value corresponding to the time, and the value can be distinguished by setting different colors. A space–time cube is formed by multiple time planes to express the change of spatial–temporal data. Its main advantage is that the spatial–temporal data can be expressed in a three-dimensional cube, highlighting the changes in geographical phenomena over time. Similar to parallel coordinates, when the amount of data is large and the attribute dimension is large, it will cause problems such as plane overlapping, path chaos, and multi-attribute challenges to represent [46]. After the data are clustered and dimensioned by SOM dimension reduction technology, the dimension of data attributes is reduced. The color can be used to represent the classification of data. With the help of geographical display tools such as maps, the expression of spatial–temporal data is completed, and the spatial–temporal attribute relationship of data after SOM dimension reduction is displayed. Thus, the visualization of high-dimensional spatial–temporal data is realized.

The SOM-based 3D dynamic display platform for food security is divided into three modules: the food data layer, mining layer, and 3D visual interface layer for food security. The food data layer stores spatial–temporal data and supports each module to call food data. The mining layer classifies food data based on SOM’s high-dimensional food data reduction and mining, providing a data basis for 3D visualization of food security [47]. The 3D visualization interface layer of food security is the platform visualization display layer, which is used for the 3D visualization expression of food data. Visualization mainly includes two kinds of 3D visualization representation before and after food data mining. The former is mainly used to retrieve food data and acquire interesting data sources through food data retrieval. In contrast, the latter is mainly used to carry out the 3D visual expression of food data based on SOM dimensionality reduction and clustering of data sources selected by the former. The main expression tools include the U-matrix algorithm, parallel coordinate, space–time cube, etc.

### 2.2. Construction of Food Security Early Warning Platform

The basis of constructing the early warning platform is constructing a relatively perfect and feasible indicator system. If the indicator is selected incorrectly, the system is not reasonable. Thus, appropriate indicators that can be used for prediction must be selected.

The economic early warning platform plays an essential role in promoting the development of the national economy [48] As an important part of the agricultural early warning system, the food security early warning platform plays an essential and positive role in ensuring food security. The purpose of studying the past is to guide future development. Therefore, the food security system must put the early warning work in an important position. Furthermore, a complete early warning system is the main work to maintain food security in the future [49,50,51].

Specifically, constructing a food security early warning platform can collect all kinds of data directly or indirectly related to food security and provide data sources for security status identification [52,53]. Through the data collection system, the platform can collect all kinds of data on food security. The data collection mainly depends on the information network of each region, which is distributed in each main producing area, main marketing area, and production and marketing area. Food security data mainly include production, consumption, circulation, grain output, climate sub-disaster situation, consumption, import and export volume, etc. Through a series of food data collection, a database can be built to prepare for identifying past and current food security situations and provide historical data material for predicting.

The early warning platform of food security is an important achievement of national informatization construction and provides a basis for scientific decision-making in the food department. The traditional food security early warning platform is mainly based on the economic cycle theory of the forecast of the prosperity index, statistical single or multiple indicators of the early warning model. The amount of data involved is limited and primarily aimed at a particular aspect of the early warning model, such as a unilateral yield forecast. The food security early warning platform proposed in this paper will be based on the original model, integrating various models to show the food security situation better. The decision system in the food security early warning platform includes the choice of model and the formulation of the decision scheme. The early warning platform is based on past data, that is, historical data, which tends to be relatively comprehensive, showing the situation of food security in the past and identifying the root causes of long-standing food insecurity problems [54,55]. The food security early warning platform can realize the observation of the development trend of long-term food security in the future. The early warning is prediction-based, which cannot only observe the prediction results but also process and describe the future short-term and long-term food security situation according to historical and realistic data. Through the investigation and analysis of the system operation status, different degrees of alarm can be issued to the possible problems so that the decision-makers can observe the problem, formulate the excluding warning plan in time, and reduce the loss caused by food insecurity. Food security early warning platforms can trigger the alarm and make various decision-making plans according to experience to facilitate relevant departments to select the best choice [56]. At the same time, it can dynamically monitor the implementation of relevant programs and assess the programs at any time. When the current programs are not suitable for the current situation, the platform will immediately give feedback to decision-makers and report the problems of the programs for better correction by decision-makers. Different regions have different food conditions. Through various quantified information, early warning models in the early warning platform are constructed to achieve more accurate forecasts and more practical food security early warning.

The food security early warning platform is based on history and is more focused on the critical pillar of the future to ensure food security. It is the only way to modernize the future food security powers. Only a high-level warning platform can promote the progress of China becoming a great nation of food security and thus contribute to world food security.

The selection of various indicators in the food security early warning platform must follow the principles of representativeness, comprehensiveness, and operability. Representativeness means that the indicators selected must be from accurate and reliable sources, closely related to the warning to be carried out. The comprehensiveness means that the selected indicators can represent the whole early warning platform, and all aspects are representative indicators. They should not be too one-sided, and the characteristics and operating rules of the system should be inferred through the selected indicators. The operability means that the selected indicators must be available, followed by statistical analysis. The data must be complete and processed by various software, rather than desultorily. Warning situation indicators refer to conditions that tend to trigger economic alarm. The determination of warning situation indicators is the most basic and vital prerequisite for early warning.

### 2.3. Data Collection

To comprehensively reflect the food security situation in China, the warning situation indicator is divided into three subsystems, namely production, consumption, circulation, and reserve, with a total of 10 indicators. The data are selected from 2007 to 2016 for 10 years, all from China Statistical Yearbook [57]. Since the food security data from 2007 to 2016 are very representative and accurate, the data span is considerable, which may cover all levels of warning intensity. Therefore, we select the food security data from 2007 to 2016 for warning intensity analysis, and the early warning indicator system data (2007–2016) is shown in Table 1. The production system includes five indicators: total output, grain planting area, the proportion of disaster area to affected area, fertilizer application amount, and irrigation area of cultivated land. The consumption system includes two indicators: the grain export volume to major agricultural products and the grain import volume to major agricultural products import volume. Circulation and reserve systems include grain self-sufficiency rate, grain reserve rate, and grain foreign trade dependence.

Therefore, In this paper, *x*_1_, *x*_2_, *x*_3_, *x*_4_, *x*_5_, *x*_6_, *x*_7_, *x*_8_, and *x*_9_ are selected as ten warning indicators to represent total output, grain planting area, proportion of disaster area to affected area, fertilizer application amount, irrigation area of cultivated land, grain export volume to major agricultural products, grain import volume to major agricultural products import volume, grain self-sufficiency rate, grain reserve rate, and grain foreign trade dependence. The analysis of warning situation indicators is an integral part of early warning. Without them, the hazard degree of warning cannot be predicted. Therefore, the selection of warning situation indicators is the key to determining the outbreak of a warning situation. In order to ensure the scientificalness of warning situation indicators, we select five suitable warning situation indicators using principal component analysis [58], namely, total output, grain planting area, irrigation area of cultivated land, the proportion of grain exports to primary agricultural products exports, and grain reserve rate, to replace the original ten indicators, which are more straightforward and more representative. After selecting the independent variable, we need to select a comprehensive dependent variable that can reflect food security [59,60].

The warning boundary refers to the threshold value divided by the warning degree. Generally, it is the upper and lower limit artificially divided by scholars through qualitative research and comprehensive data [61]. It is also called a threshold value. The division of warning boundary is the core to determine the size of warning intensity, according to different thresholds to determine all kinds of warning intensity. Warning intensity refers to the outbreak of warning used to judge the intensity of the warning situation. The setting of warning intensity should be based on the constantly changing state of warning situation indicators. More importantly, different standards are divided according to the threshold value into different periods to analyze the current situation in a specific warning boundary and then predict the warning situation. The warning intensity is determined as four levels: no warning, light warning, medium warning, and heavy warning, respectively, with the numbers 0, −1, −2, and −3 representing their degree. According to the operation status of the grain market, the relevant warning intensity degree is defined, as shown in Table 2. Next, the upper and lower warning situation thresholds are determined to divide the actual range of each warning intensity level. The determination of warning intensity is generally set artificially. This paper selects reasonable warning boundaries to judge China’s food security according to international standards, the studies of many scholars in the past, and the current economic situation in China, and the setting of warning boundaries is strict [52,62].

As an essential branch of support vector machine (SVM), support vector regression (SVR) is used to solve regression problems and is often used to predict the weather, stocks, and other aspects [63,64]. When SVM is used to process data, it is often assumed that the training samples are linearly separable, which can be solved by the above method. However, if the training samples cannot find a hyperplane that can be divided, the kernel function can be introduced to solve the problem. SVR can solve the problem of linear inseparability through the intervention of the kernel function. Regarding how to select kernel functions, certain kernel functions can be selected according to the experience and research of experts and scholars, or cross-validation can be adopted to test the test results of different kernel functions. If the results of the above methods are not ideal, the method of mixed kernel function can also be adapted to combine different kernel functions and establish new kernel functions. However, the complexity of calculation increases, so this paper does not adopt the mixed form and only selects different kernel functions for model analysis.

Regarding total output, the standard to determine whether the total output is within a reasonable range is calculated according to the population growth. If the grain output can meet the grain demand of everyone, the ideal grain output can be obtained by multiplying the per capita grain output with the total national population, and then the security of the total grain output can be judged. The population of the country is from China Statistical Yearbook 2021. Internationally, it is generally believed that the per capita food possession reaches 400 kg, that is, it reaches the basic level. However, the food conditions in China are improving in 2020. Therefore, 400 kg per capita food possession is set as the lower limit of no warning. China used to take the per capita grain possession of 300 kg and 200 kg as the evaluation standard of having only basic needs, as early as in the 1990s, the per capita grain possession of China exceeded 300 kg. From this point of view, it is more reasonable to set 375 kg and 350 kg of grain per capita as the lower limit of light warning and the upper limit of heavy warning.

Regarding grain planting area, the criterion is to calculate the standard proportion of the planting area of grain to the total planting area of crops and then multiply this value with the total planting area of crops. The total planting area of crops is from China Statistical Yearbook 2021, and the ratio is generally between 0.6 and 0.7. In order to refine this indicator and more dynamically reflect the state of food security, the lower limit of the proportion of no warning is set as 0.68, the lower limit of the proportion of light warning is set as 0.67, and the lower limit of the proportion of medium warning is set as 0.66.

Regarding the irrigation area of cultivated land, the criterion is generally the product of the standard proportion of irrigation area to cultivated land area and cultivated land area. Considering the current grain irrigation situation, the preliminary judgment of the proportion is generally between 0.04 and 0.05. Within a reasonable range, the higher the proportion of irrigated area, the better the reduction of grain production caused by drought and other disasters. In order to better reflect the precise influence of irrigation on grain, the proportion of 0.048, 0.045, and 0.042 close to 0.05 and 0.04 are used as the proportion threshold of each warning intensity, which is multiplied by the cultivated land area to obtain the standard value of the warning boundary under the four warning intensities.

Regarding grain export volume to major agricultural products, the criterion is based on the proportion of grain foreign trade dependence. It is generally believed that the higher the proportion, the lower the food security. Therefore, the range of no warning is (−∞, 0.7], light warning is (0.7, 0.8], medium warning is (0.8, 0.9), and heavy warning is (0.9, +∞).

Regarding the grain reserve rate, according to the international standard, the grain reserve rate between 0.17 and 0.18 reaches the level of food security. Only when the grain reserve rate is lower than 0.17, it is considered to be lower than the safety line. When it is below 0.14, it is a food security crisis. However, this international standard is not applicable in China. In this paper, the grain reserve rate will be combined with China’s grain import and export and inventory status, and the threshold will be set as the following standard. The range of no warning is (0.8, +∞), the range of light warning is (0.6, 0.8), the range of medium warning is (0.4, 0.6), and the range of heavy warning is (−∞, 0.4).

After determining the warning boundary of different warning intensities, the criterion of warning intensity is formed, and then it is necessary to carry on the next detailed analysis of the warning intensity. In the early warning system of this paper, a preset provision is made, that is, when the warning situation is no warning and light warning, no warning will be triggered. When medium warning and heavy warning occur, the warning is triggered. It is important to identify all the warning indicators and determine the warning situation indicators that cause the warning. It is very important to find the warning source quickly. Only by finding out the source can we clear the dangerous situation in time and formulate the relevant detailed strategy, so as to truly alleviate the crisis. Early warning systems formulate the rules of the warning trigger mechanism, which is equivalent to the function of a switch, according to whether to pass the switch or to implement different strategic arrangements. Through the above calculation, the upper and lower thresholds of each warning intensity of food security can be preliminarily determined. By substituting the data of each indicator in each year into the threshold, the security degree of each indicator in each year can be determined, as shown in Table 3.

As seen in Table 3, from 2008 to 2014, the increase in planting area and other factors led to the rapid growth of total output, which made the two warning intensities quickly turn into no alarm state. At the same time, the irrigated area showed different degrees of light warning and medium warning states, mainly from 2009 to 2010. The warning intensity of grain export volume compared to the export volume of primary agricultural products showed a declining trend. After 2009, a light warning state appeared that did not need a warning, while the security degree of grain reserve rate mainly showed a medium warning state. In 2013, the warning intensity showed a decreasing trend and rapidly entered a light warning state and no warning state. The improvement of the security degree of these two indicators mainly depends on the change in grain collection and storage policy. In 2015, there was no warning state; only the planting area was lightly warned, while all other indicators showed no warning signs. In 2016, there was no warning, in which the planting area was a light warning, and the proportion of grain export volume in the export volume of major agricultural products was a medium warning. Since other indicators were in no warning state, which could make up for the warning situation in export, it was consistent with the fact that there was no warning state this year.

The SVR-based food security early warning platform divides the samples into training and test sets. The training set is used to simulate data. In contrast, the test set is used to test whether the model presented by the training set has a solid explanatory ability to judge whether the model is adequate. The training set and test set proportion is 80% and 20%, respectively. The model has ten samples and five features. Therefore, ten samples of the training set and three samples of the test set were determined. The training samples were input into the SVR model to adjust the parameters and determine the optimal model. Specific parameter settings are shown in Table 4.

## 3. Results

In SVR model, y_true represents the true value, and y_pre_kernel represents the predicted value under different kernel functions. When epsilon = 0.1 and other parameters remain unchanged by default, the predicted results are shown in Table 5. As seen from Table 5, in the case of epsilon = 0.1, when the kernel function is rbf, the predicted value in each year is 1.6238, which differs significantly from the true value without any fluctuation. In the case of linear and poly, the values predicted by the linear kernel function are closer to the true values than those predicted by the poly kernel function. Generally, when epsilon is enlarged, the model’s prediction accuracy will be improved. When epsilon is enlarged to a certain extent, no matter how much epsilon is enlarged, the prediction result will not be affected. Therefore, selecting an appropriate epsilon is necessary, defining epsilon as 1.

The predicted results are shown in Table 6. As seen from Table 6, when epsilon is increased, not only can the prediction result not be more accurate, but also the predicted value of the three kernel functions deviates further from the true value. Moreover, the predicted value of the three kernel functions each year is 1.6961, and the true value of the three years is more than 2. Obviously, this model is not applicable, and epsilon cannot be defined as 1. Therefore, epsilon is defined as 0.2 in the parameter adjustment, and the predicted results are shown in Table 7. It can be seen from Table 7 that the predicted value of the linear kernel function is very close to the true value, while the predicted value of the rbf kernel function is quite different from the true value. The predicted value of the poly kernel function is also close to the true value but not as good as the linear fitting. Therefore, the linear kernel is the most suitable of the three.

Since the linear kernel is the kernel with the best performance in the training set and test set results, and it does not need to adjust other parameters, the current parameters are optimal. The results of the prediction model made by linear kernel function are put into the warning boundary, and the warning intensity of the average selling price of major grains per kilogram and the warning intensity predicted from 2014 to 2016 is obtained, as shown in Table 8. There is almost no difference between the alarming degree of predicted value and the real warning intensity, which indicates that the SVR model is very effective for food security early warning, which is very consistent with the actual food security operation state, which also indicates that the food security early warning platform based on SVR is very reasonable.

## 4. Discussion

By constructing a 3D dynamic display and early warning platform for food security, we can intuitively understand the real-time situation of food, which is of great significance in promoting the development of agriculture [65,66]. Ensuring food security is the foundation of economic development and social stability. Currently, COVID-19, extreme weather, and other uncertain events have brought a chain reaction to the global food industry chain and supply chain, and the overall situation of the international food market is not optimistic, raising higher requirements for China to ensure food security and participate in global food governance effectively. Next, we will study China’s medium and long-term food security strategy.

In the face of the current international food situation and new developments, to ensure domestic food security, we must attach great importance to and solve the short-term impact of the international market. First, strengthen early warning and monitoring of domestic and international grain markets, effectively guide and manage expectations, and ensure unimpeded information, transportation, and logistics in domestic grain markets and circulation. Second, we will stabilize grain yields through cost-cutting and supplementary measures, release some reserves of potash fertilizer and fertilizer for disaster relief ahead of schedule, and curb the rapid rise in domestic fertilizer prices. In response to the rapidly rising prices of agricultural supplies, we provided one-time subsidies to farmers who grew grain and promoted the establishment of a system of subsidies for actual grain production in significant rice and wheat-producing areas. Third, we will improve the domestic grain emergency supply system. Depending on the scale of emergency grain reserves, we will meet the needs of large and medium-sized cities for grain consumption for 10 to 15 days and speed up the construction of emergency grain processing enterprises, emergency supply outlets, emergency storage and transportation enterprises, and regional distribution centers.

As bulk agricultural products, the market competitiveness of grain depends on grain price. In recent years, China’s minimum purchase price policy for rice and wheat has played an essential role in ensuring grain production capacity and stabilizing the income of food and agriculture. However, the grain price mechanism has not been fundamentally adjusted, resulting in an insufficient supply of high-quality grain. To this end, we should continue to adhere to the direction of grain market reform, and the minimum purchase price policy of grain should not be changed in the short term as long as it conforms to the WTO micro allowance rules after limited purchase and storage. In order to reduce the excess ration stocks, we should strengthen the market function to regulate the output, improve the social tolerance of reasonable fluctuation of ration prices, and expand the reasonable fluctuation range of ration prices. In the long run, we will adhere to the principle of market pricing and separation of prices from subsidies to improve the mechanism for setting grain prices.

Currently, grain production in the water-rich south of China is decreasing while that in the water-deficient north is increasing. The risk of mismatch of water, soil, light, and heat resources in food production is becoming increasingly prominent. In terms of the proportion of grain output in the three-year average from 2006 to 2008 and 2016 to 2018, the share of the seven central grain-producing provinces in northern China increased from 60.3 percent to 63.9 percent, while that of the six major grain-producing provinces in southern China decreased from 39.7 percent to 37.1 percent, widening the gap by 3.6 percentage points over the past decade. By 2035, the proportion of major grain-producing areas in the north will approach 70 percent, and that in the south will approach 30 percent. At the same time, as the population continues to gather in critical regions and cities, especially in recent years, the trend of population migration from north to south is noticeable, and food consumption and security risks are also concentrated in these regions. In a state of emergency and under national logistics constraints, food supplies may be strained in some regions if food is transported externally. Therefore, we must attach great importance to the responsibility of food security in urban agglomerations and metropolitan areas and take adequate measures to deal with it.

In the coming period, China’s agricultural production mode will be changed entirely, the elderly farmers will gradually withdraw from food production, the new agricultural operation subject will gradually grow, and the main body of grain-growing will experience a replacement process. However, the overall progress of the rural land system reform in China is slow, the grain price mechanism has not been fundamentally straightened out, the income of grain growing is not high, resulting in moderate scale operation, and scientific and technological grain growing is restricted. The cultivation of new business subjects is not fast. Based on this, considering China’s primary national conditions and agricultural conditions, we should speed up the improvement of the agricultural production service system, strengthen the construction of interest linkage mechanism, introduce a large number of small farmers into the track of modern agricultural development through effective forms such as land trusteeship, promote moderate scale operation of agriculture, and promote the joint development of family operation and cooperative operation, enterprise operation and collective operation based on grain subject dominated by family operation. At the same time, we will accelerate the reform of the rural land and collective property rights systems, strengthen the application of modern information technology, comprehensively upgrade equipment for whole-process mechanized production, and promote mechanized grain growth.

## 5. Conclusions

Ensuring critical agricultural products, especially food security, is the foundation of national economic and social stability and development. First, this paper combines the clustering dimension reduction algorithm with several other visualization methods to realize the 3D visualization of food security with multi-window collaboration, overcomes the problems of dimension and sample size limitations existing in a single visualization method, and dramatically improves the efficiency of mining, provides new ideas for analysis and mining of multi-dimensional spatial–temporal data, and can provide proper technical support for data mining and analysis of mass agriculture. It benefits the development and promotion of fine agriculture and has specific economic and social benefits. However, different clustering methods have different abilities to fit the topological characteristics of datasets, leading to differences in clustering accuracy. Different visualization methods have different emphases on data visualization expression. In this regard, for different agricultural datasets, finding a suitable clustering dimensionality reduction algorithm, determining its topological distribution, judging its clustering accuracy, and choosing appropriate visual tools to display its spatial–temporal relationship is worth further research. Then, we expound on how to construct the food security early warning platform. Among them, the indicator to select the right is a top priority. We select total output, grain planting area, irrigation area of cultivated land, the proportion of grain exports to major agricultural product exports, and grain reserve rate to determine four warnings: no warning, light warning, medium warning, and heavy warning. The relevant scope is determined through careful division of warning boundaries, trigger rules are formulated, and alarm measures are taken for medium and heavy warnings that trigger alarms. Finally, the SVR model was used to predict China’s food security situation from 2014 to 2016, and it was found that the predicted value was not significantly different from the true value. The warning intensity is consistent with the actual situation, which shows that the food security early warning platform is effective.

## Figures and Tables

**Figure 1 ijerph-19-11169-f001:**
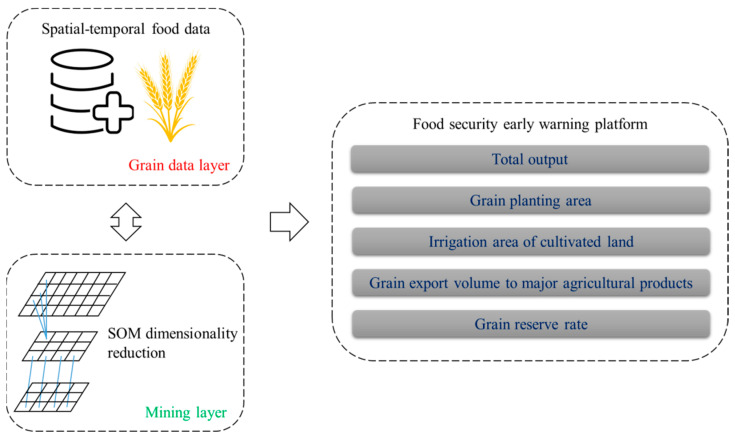
The overall framework of SVR-based food security early warning platform.

**Figure 2 ijerph-19-11169-f002:**
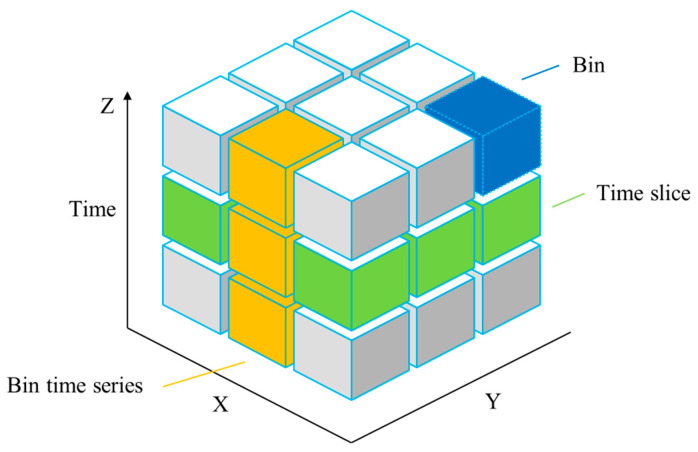
Space–time cube schematic diagram.

**Table 1 ijerph-19-11169-t001:** Early warning indicator system data (2007–2016).

	Total Output ^a^	Grain Planting Area ^b^	Proportion of Disaster Area to Affected Area ^c^	Fertilizer Application Amount ^d^	Irrigation Area of Cultivated Land ^e^	Grain Export Volume to Major Agricultural Products ^f^	Grain Import Volume to Major Agricultural Products Import Volume ^g^	Grain Self-Sufficiency Rate ^h^	Grain Reserve Rate ^i^	Grain Foreign Trade Dependence ^j^
2007	504.13	105,638	51.2	51.08	56,518	0.947811	0.716608	0.942773	0.415082	0.064394
2008	534.34	106,793	55.7	52.39	58,472	0.822455	0.762684	0.949857	0.499551	0.074022
2009	539.40	108,986	45.0	54.04	59,261	0.824163	0.780543	0.918906	0.509659	0.082461
2010	559.11	109,876	49.5	55.62	60,348	0.775617	0.811473	0.916366	0.475717	0.105588
2011	588.49	110,573	38.3	57.04	61,682	0.781934	0.787945	0.904983	0.483806	0.096839
2012	612.22	111,205	46.0	58.39	62,491	0.791962	0.782386	0.89829	0.526525	0.115049
2013	630.48	111,956	45.6	59.12	63,473	0.778135	0.792837	0.854517	0.666741	0.114422
2014	639.64	112,723	50.9	59.96	64,540	0.706001	0.86811	0.692614	0.937113	0.09834
2015	660.60	113,343	56.9	60.23	65,873	0.614453	0.873351	0.65102	1.128235	0.105107
2016	660.43	113,034	52.1	59.84	67,149	0.819672	0.901062	0.5953	1.164821	0.09852

Notes: a, b, c, d, e, f, g, h, i, and j denote the units of warning indicators. ^a^ Mt; ^b^ kha; ^c^ %; ^d^ Mt; ^e^ kha; ^f^ %; ^g^ %; ^h^ %; ^i^ %; ^j^ %.

**Table 2 ijerph-19-11169-t002:** Warning intensity definition.

	No Warning (0)	Light Warning (−1)	Medium Warning (−2)	Heavy Warning (−3)
Food security level	Good	Relatively good	Relatively poor	Poor
Food market conditions	Supply–demand balance	Basically supply–demand balance	Basically supply–demand imbalance	Totally unbalanced
Price increase	Rational	Acceptable	Governable	Uncontrollable

**Table 3 ijerph-19-11169-t003:** Early warning level of food security in 2007–2016.

Year	Total Output	Grain Planting Area	Irrigation Area of Cultivated Land	Proportion of Grain Exports to Major Agricultural Products Exports	Grain Reserve Rate	Average Selling Price per Kilogram of Staple Grain
2007	−1	0	−1	−3	−2	−2
2008	0	0	0	−2	−2	−2
2009	0	0	−2	−2	−2	−2
2010	0	0	−2	−1	−2	−2
2011	0	0	−1	−1	−2	−2
2012	0	0	−1	−1	−2	−2
2013	0	0	−1	−1	−1	−2
2014	0	0	−1	−1	0	−2
2015	0	−1	0	0	0	0
2016	0	−1	0	−2	0	0

Notes: 0: no warning; −1: light warning; −2: medium warning; −3: heavy warning.

**Table 4 ijerph-19-11169-t004:** Parameters setting.

Parameter	Setting
Penalty factor	1
Cache size	200 M
Kernel function	linear/poly/rbf
Shrinking	TRUE
Verbose	FALSE
Epsilon	0.1
Coef0	0
Gamma	auto deprecated
Tol	0.001

**Table 5 ijerph-19-11169-t005:** SVR predicted results (epsilon = 0.1).

Year	y_true	y_pre_Linear	y_pre_rbf	y_pre_Poly
2014	2.4678	2.4791	1.6238	2.5415
2015	2.3562	2.6657	1.6238	2.7634
2016	2.2768	2.6444	1.6238	2.8312

**Table 6 ijerph-19-11169-t006:** SVR predicted results (epsilon = 1).

Year	y_true	y_pre_Linear	y_pre_rbf	y_pre_Poly
2014	2.4678	1.6961	1.6961	1.6961
2015	2.3562	1.6961	1.6961	1.6961
2016	2.2768	1.6961	1.6961	1.6961

**Table 7 ijerph-19-11169-t007:** SVR predicted results (epsilon = 0.2).

Year	y_true	y_pre_Linear	y_pre_rbf	y_pre_Poly
2014	2.4678	2.4687	1.6508	2.5808
2015	2.3562	2.6832	1.6508	2.7241
2016	2.2768	2.4416	1.6508	2.7109

**Table 8 ijerph-19-11169-t008:** The warning intensity of predicted and true values.

Year	Warning Intensity of Predicted Value	Warning Intensity of True Value
2014	−2	−2
2015	0	0
2016	0	0

## Data Availability

All data used to support the findings of the study are included within the paper.

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
