# Peer review of "Food Security: 3D Dynamic Display and Early Warning Platform Construction and Security Strategy"

_ijerph, 2022, doi:10.3390/ijerph191811169_

Round 1

Reviewer 1 Report

Summary: It was a pleasure reviewing this paper. IJERPH readers would find the topic interesting. It is clear how the manuscript is structured and written. There are a few points that could be improved in the study.

Point 1: The abstract should be revised to be more concise and reflect the results of the study. For example, what are the outcomes of the study? Solutions and suggestions? As explained in the abstract, the authors just described their process of writing and completing the article.

Point 2: Line no. 105 and 106 The author needs to clarify the meaning.

Point 3: In Figure 1. doesn’t have supplementary data. How authors develop this figure? What is the data source? I suggest writing more explanations after “Figure 1. Space-time cube diagramNotes:…….. “.

Point 4: In Table 1. For better understanding, I suggest adding notes explaining the figure in the table. E.g. meaning of 0, -1, -2,-3? 

Author Response

All modifications have been marked in Red color.

Point 1: The abstract should be revised to be more concise and reflect the results of the study. For example, what are the outcomes of the study? Solutions and suggestions? As explained in the abstract, the authors just described their process of writing and completing the article.

Response: Thanks for this comment. We reshape the abstract, removing some redundant descriptions to make it appear more concise. Additionally, we provide illustrations of the experiment results and appropriate recommendations. The revised part is highlighted in red on lines 11 - 26.

Point 2: Line no. 105 and 106 The author needs to clarify the meaning.

Response: We reinterpret the meaning of lines 105 and 106. The revised part is highlighted in red on lines 148 - 152.

Point 3: In Figure 1. doesn’t have supplementary data. How authors develop this figure? What is the data source? I suggest writing more explanations after “Figure 1. Space-time cube diagramNotes:…….. “.

Response: For a more precise representation of the schematic diagram of a space-time cube, we reconstruct the previous Figure 1, called Figure 2 in the revised version, from a different perspective. Besides, we explain further the schematic diagram of the space-time cube. The revised part is highlighted in red on lines 170 – 178 and Figure 2.

Point 4: In Table 1. For better understanding, I suggest adding notes explaining the figure in the table. E.g. meaning of 0, -1, -2,-3? 

Response: The paper contains the interpretation of warning intensity (0, -1, -2, -3), which is not conducive to the reader's search. To this end, we add notes to Table 3 for explaining the ambiguous 0, -1, -2, -3. The revised part is highlighted in red on line 402.

Reviewer 2 Report

Comment: Thank you for the opportunity to review the manuscript entitled “Food Security: 3D Dynamic Display and Early Warning Platform Construction and Security Strategy”, submitted for publication to the International Journal of Environmental Research and Public Health. The manuscript is interesting and original. However, it should be structured according to the Instruction for Authors provided by MDPI, namely: (i) Introduction; (ii) Materials and Methods; (iii) Results; (iv) Discussion; (v) Conclusions. Overall, it is not clear in its structure and it appears confusing and not easy to read. Moreover, several references are missing either in the section “Introduction”, “Materials and Methods” and “Literature review”. The authors must enhance the scientific soundness of their research and 

Abstract: The abstract is comprehensive. It accounts for approx. 200 words and is in line with the Instruction for Authors provided by MDPI. 

General comment: Please, write references as suggested by the Instruction for Authors. For instance, line 30, it should be [1,2] instead of [1-2]. Please, revise references also in the reference list. 

Introduction:

Please, add more references to justify the statements provided in the section “Introduction”. For instance, lines 30-32 could cite the most recent studies on the environmental, social and economic issues related to grain production, such as: 

Bux, C., Varese, E., Lombardi, M., Amicarelli, V. (2022). Economic and Environmental Assessment of Conventional versus Organic Durum Wheat Production in the Mediterranean Area. Sustainability, 14(15), 9143. https://doi.org/10.3390/su14159143

The authors must declare (and justify) from the very beginning that their research applied the proposed tools to food security in the grain production, and highlight the role of “grain production” towards the enhancement of food security. 

Moreover, when dealing with “food security” – I expect reading the official definition provided by the World Food Summit (1996), namely “all people, at all times, have physical and economic access to sufficient, safe and nutritious food that meets their dietary needs and food preferences for an active and healthy life” – please add some more references. I would introduce one of the main challenges towards the enhancement of food security, namely the reduction of food waste (as proposed by the Sustainable Development Goals introduced by the United Nations). As regards to food waste minimization, please refer to:  

World Food Summit (1996), Report of the World Food Summit, 13-17 November 1996, FAO, Rome.

Amicarelli, V., Roe, B.E., Bux, C. (2022).  Measuring the Economic Costs of Food Loss and Waste in the Italian Potato Chip Industry using Material Flow Cost Accounting. Agriculture, 12, 523. https://doi.org/10.3390/ agriculture12040523

Poore, J., Nemecek, T. (2018). Reducing food’s environmental impacts through producers and consumers. Science 2018, 987–992. 

Lines 39-40. Please, provide data (statistics and figures) related to the food security issue on the global and the local level. Consider reading the official reports provided by FAO.

Line 45. “661,6072 million tons” is written in a wrong manner. Further, use acronyms when referring to million tons (Mt). Also at line 47, “960000 hectares” is written in a wrong manner, as well as at line 48. Please, use a common unit of measurement, such as Mt, instead of tons, as provided at line 48 (i.e., “65789200 tons”). Further, lines 42-49 must be supported by references and official statistics. 

Lines 51-61. The authors introduce the topic of the “Internet of Things” “remote sensing technology”, etc. Please, provide some more details related to the benefits introduced by such innovative tools in the agri-food sector. Consider referring to the subsequent (and recent) articles on the topic. 

Rana, R.L.Tricase, C. and De Cesare, L. (2021). Blockchain technology for a sustainable agri-food supply chain. British Food Journal, Vol. 123 No. 11, pp. 3471-3485. https://doi.org/10.1108/BFJ-09-2020-0832

Bux, C., Varese, E., Amicarelli, V., Lombardi, M. (2022). Halal Food Sustainability between Certification and Blockchain: A Review. Sustainability, 14(4), 2152. https://doi.org/10.3390/su14042152

Lines 89-92. I would give more emphasis to the “contributions of this paper”, for instance in a specific paragraph, which sums up all the previous theoretical assumptions. 

SOM-Based Collaborative Multi-View 3D Dynamic Visualization of Food Security: I understand the present heading as a “Theoretical background” or “Literature review” section. Please, add more references related to previous studies applying similar methods towards the enhancement of food security, and highlight the main outcomes, as to strengthen the originality of the present research compared to previous studies. 

Construction of Food Security Early Warning Platform Based on SVR: Is the present section devoted to the description of the “Materials and Methods” applied in the present research? Please, according to the Instruction for Authors provided by the Journal, you should follow the subsequent research article structure: (i) Introduction; (ii) Materials and Methods; (iii) Results; (iv) Discussion; (v) Conclusions. It is important to help readers understand the research and make the study consistent. 

Overall, before describing in detail the “Materials and Methods” applied in the research, at the beginning of such paragraph the authors must provide a brief overview of the research framework. They could do it also providing a clear and comprehensive Figure. 

Line 153-154. The new model will be based on the “original model”. Which is the original model?

Lines 129-180. It seems to me that such lines must include more references. Specifically, when the authors declare that the new food security early warning platform “is based on past data, that is, historical data, which tends to be relatively comprehensive, showing the situation of food security in the past and identifying the root causes of long-standing food insecurity problems”, I expect the authors justifying such (strong) assumption. The authors must refer to previous studies and enhance the scientific soundness of their research. 

Lines 194-198. What do you think about including the ten warning indicators within a Table? It could highlight the selected variables. Moreover, on the basis of which criteria have you selected such variables? I expect an in-depth dissertation related to their “representativeness, comprehensiveness and operability” (lines 183) in China. 

Also, lines 218-220, which are essential to understand the subsequent results, should be better clarified. Further, how warning intensity is determined within four levels? According to which criteria or which previous research they are identified with 0, -1, -2, -3 (line 220)?

Lines 223-225.Which international standards? Which studies of many scholars? Please, provide details. Such a description is rather insufficient. Detailed references related to the data sources are required. 

Overall, the sub-heading “data collection” is missing. Where data have been retrieved? Please, provide in-depth details related to the data sourced. 

Table 1. It is not clear how such levels have been calculated.

I suppose that lines 274-310 should be included in the section “Materials and Methods”, whereas lines 311-337, which provide “predicted results”, should be included in the section “Results”. 

“Medium and Long-Term Food Security Strategy” could be entitled “Discussion”. 

Author Response

All modifications have been marked in Blue color.

  1. Comment: Thank you for the opportunity to review the manuscript entitled “Food Security: 3D Dynamic Display and Early Warning Platform Construction and Security Strategy”, submitted for publication to the International Journal of Environmental Research and Public Health. The manuscript is interesting and original. However, it should be structured according to the Instruction for Authors provided by MDPI, namely: (i) Introduction; (ii) Materials and Methods; (iii) Results; (iv) Discussion; (v) Conclusions. Overall, it is not clear in its structure and it appears confusing and not easy to read. Moreover, several references are missing either in the section “Introduction”, “Materials and Methods” and “Literature review”. The authors must enhance the scientific soundness of their research and 

Response: Thanks for this comment. According to the Instruction for Authors provided by MDPI, we rearrange the structure of the article. Accordingly, we revise of the structure of the paper. The revised part is highlighted in blue on lines 127 - 130.

  1. General comment: Please, write references as suggested by the Instruction for Authors. For instance, line 30, it should be [1,2] instead of [1-2]. Please, revise references also in the reference list. 

Response: We revise reference format.

  1. Please, add more references to justify the statements provided in the section “Introduction”. For instance, lines 30-32 could cite the most recent studies on the environmental, social and economic issues related to grain production, such as: 

 Bux, C., Varese, E., Lombardi, M., Amicarelli, V. (2022). Economic and Environmental Assessment of Conventional versus Organic Durum Wheat Production in the Mediterranean Area. Sustainability, 14(15), 9143. https://doi.org/10.3390/su14159143

Response: We add more references to illustrate the environmental, social and economic issues related to grain production. The revised part is highlighted in blue on line 34.

  1. The authors must declare (and justify) from the very beginning that their research applied the proposed tools to food security in the grain production, and highlight the role of “grain production” towards the enhancement of food security.

Response: At the end of the first paragraph of the Introduction section, we state that a machine learning-based warning platform can be applied to food security and highlight the role of grain production in enhancing food security. The revised part is highlighted in blue on lines 51 - 62.

  1. Moreover, when dealing with “food security” – I expect reading the official definition provided by the World Food Summit (1996), namely “all people, at all times, have physical and economic access to sufficient, safe and nutritious food that meets their dietary needs and food preferences for an active and healthy life” – please add some more references. I would introduce one of the main challenges towards the enhancement of food security, namely the reduction of food waste (as proposed by the Sustainable Development Goals introduced by the United Nations). As regards to food waste minimization, please refer to:  

World Food Summit (1996), Report of the World Food Summit, 13-17 November 1996, FAO, Rome.

Amicarelli, V., Roe, B.E., Bux, C. (2022).  Measuring the Economic Costs of Food Loss and Waste in the Italian Potato Chip Industry using Material Flow Cost Accounting. Agriculture, 12, 523. https://doi.org/10.3390/ agriculture12040523

Poore, J., Nemecek, T. (2018). Reducing food’s environmental impacts through producers and consumers. Science 2018, 987–992. 

Response: According to the FAO, we update the definition of food security. Moreover, one of the main challenges in enhancing food security is to reduce food waste. For each point, we add references. The revised part is highlighted in blue on lines 37 – 40 and lines 44 – 51.

  1. Lines 39-40. Please, provide data (statistics and figures) related to the food security issue on the global and the local level. Consider reading the official reports provided by FAO.

Response: We provide data related to the food security issue on the global and the local level by considering the official reports provided by FAO. The revised part is highlighted in blue on lines 64 - 70.

  1. Line 45. “661,6072 million tons” is written in a wrong manner. Further, use acronyms when referring to million tons (Mt). Also at line 47, “960000 hectares” is written in a wrong manner, as well as at line 48. Please, use a common unit of measurement, such as Mt, instead of tons, as provided at line 48 (i.e., “65789200 tons”). Further, lines 42-49 must be supported by references and official statistics. 

Response: We revise the unit of measurement and add references. The revised part is highlighted in blue on lines 72 - 79.

  1. Lines 51-61. The authors introduce the topic of the “Internet of Things” “remote sensing technology”, etc. Please, provide some more details related to the benefits introduced by such innovative tools in the agri-food sector. Consider referring to the subsequent (and recent) articles on the topic.

Rana, R.L., Tricase, C. and De Cesare, L. (2021). Blockchain technology for a sustainable agri-food supply chain. British Food Journal, Vol. 123 No. 11, pp. 3471-3485. https://doi.org/10.1108/BFJ-09-2020-0832

 Bux, C., Varese, E., Amicarelli, V., Lombardi, M. (2022). Halal Food Sustainability between Certification and Blockchain: A Review. Sustainability, 14(4), 2152. https://doi.org/10.3390/su14042152

Response: We provide the benefits of these innovative tools to the agri-food sector for the Internet of Things and remote sensing technology. The revised part is highlighted in blue on lines 87 - 92.

  1. Lines 89-92. I would give more emphasis to the “contributions of this paper”, for instance in a specific paragraph, which sums up all the previous theoretical assumptions. 

Response: We have a specific paragraph for the “contributions of this paper”. The revised part is highlighted in blue on lines 123 - 126.

  1. SOM-Based Collaborative Multi-View 3D Dynamic Visualization of Food Security:I understand the present heading as a “Theoretical background” or “Literature review” section. Please, add more references related to previous studies applying similar methods towards the enhancement of food security, and highlight the main outcomes, as to strengthen the originality of the present research compared to previous studies. 

Response: We change the original Section 2, SOM-based Collaborative Multi-View 3D Dynamic Visualization of Food Security, into the present Section 2.1 Theoretical Background. Additionally, we cite some literatures for enhancement of food security. The revised part is highlighted in blue on lines 158 - 169.

  1. Construction of Food Security Early Warning Platform Based on SVR: Is the present section devoted to the description of the “Materials and Methods” applied in the present research? Please, according to the Instruction for Authors provided by the Journal, you should follow the subsequent research article structure: (i) Introduction; (ii) Materials and Methods; (iii) Results; (iv) Discussion; (v) Conclusions. It is important to help readers understand the research and make the study consistent. 

Response: We classify it to Section 2: Materials and Methods to help readers understand the research and make the study consistent.

  1. Overall, before describing in detail the “Materials and Methods” applied in the research, at the beginning of such paragraph the authors must provide a brief overview of the research framework. They could do it also providing a clear and comprehensive Figure. 

Response: In “Materials and Methods” Section, we provide a brief overview of the research framework, as well as the comprehensive Figure. The revised part is highlighted in blue on lines 132 – 138 and Figure 1.

  1. Line 153-154. The new model will be based on the “original model”. Which is the original model?

Response: We explain the original warning model and analyze the shortcomings of the conventional model. The revised part is highlighted in blue on lines 227 - 231.

  1. Lines 129-180. It seems to me that such lines must include more references. Specifically, when the authors declare that the new food security early warning platform “is based on past data, that is, historical data, which tends to be relatively comprehensive, showing the situation of food security in the past and identifying the root causes of long-standing food insecurity problems”, I expect the authors justifying such (strong) assumption. The authors must refer to previous studies and enhance the scientific soundness of their research. 

Response: We add some references for enhancing the scientific soundness of our research.

  1. Lines 194-198. What do you think about including the ten warning indicators within a Table? It could highlight the selected variables. Moreover, on the basis of which criteria have you selected such variables? I expect an in-depth dissertation related to their “representativeness, comprehensiveness and operability” (lines 183) in China. 

Response: To comprehensively reflect the food security situation in China, the food security data from 2007 to 2016 are highly representative and accurate, so we choose the data of these ten years as the research object. Furthermore, we explain the basis for selecting these warning indicators. The revised part is highlighted in blue on lines 272 – 284, Table 1, and lines 297 – 302.

  1. Also, lines 218-220, which are essential to understand the subsequent results, should be better clarified. Further, how warning intensity is determined within four levels? According to which criteria or which previous research they are identified with 0, -1, -2, -3 (line 220)?

Response: We give the definition of four warning intensity. According to the food security level, food market conditions, and price increase, the table is made to intuitively reflect the warning intensity definition. The revised part is highlighted in blue on lines 315 – 317 and Table 2.

  1. Lines 223-225.Which international standards? Which studies of many scholars? Please, provide details. Such a description is rather insufficient. Detailed references related to the data sources are required.

Response: We add some references.

  1. Overall, the sub-heading “data collection” is missing. Where data have been retrieved? Please, provide in-depth details related to the data sourced. 

Response: We add the sub-heading “data collection”. The data in Table 1 are from the China Statistical Yearbook.

  1. Table 1. It is not clear how such levels have been calculated.

Response: We show how to determine the warning intensity of total output, grain planting area, irrigation area of cultivated land, grain export volume to major agricultural products, and grain reserve rate. The revised part is highlighted in blue on lines 341 – 398.

  1. I suppose that lines 274-310 should be included in the section “Materials and Methods”, whereas lines 311-337, which provide “predicted results”, should be included in the section “Results”. 

“Medium and Long-Term Food Security Strategy” could be entitled “Discussion”.

Response: We rearrange the structure of the article according to the Instruction for Authors provided by MDPI.

We polish the paper once more and make some changes in addition to the reviewers' comments in order to improve the readability of this paper.

Round 2

Reviewer 2 Report

Comment: Thank you for the opportunity to review the revised version of the manuscript entitled "Food Security: 3D Dynamic Display and Early Warning Platform Construction and Security Strategy". The authors have substantially revised the manuscript according to the reviewer's suggestions, and have reached a suitable scientific soundness. The research is significant and original, and the quality of presentation is clear and comprehensive. References have been substantially improved, as well as the section "Materials and Methods". I can confidentially suggest the acceptance of the manuscript in the International Journal of Environmental Research and Public Health. 

I have only one concern, which must be revised in the manuscript: Table 1 must provide the unit related to the differentially warning indicators. If possibile, it could be added in the Notes of the table. 

Thank you for the great work. 

Author Response

Comment: I have only one concern, which must be revised in the manuscript: Table 1 must provide the unit related to the differentially warning indicators. If possible, it could be added in the Notes of the table.

Response: We have addressed this issue via added the notes for Table 1. Please see page 7 for details.